# Resistance to Ceftazidime/Avibactam, Meropenem/Vaborbactam and Imipenem/Relebactam in Gram-Negative MDR Bacilli: Molecular Mechanisms and Susceptibility Testing

**DOI:** 10.3390/antibiotics11050628

**Published:** 2022-05-06

**Authors:** Paolo Gaibani, Tommaso Giani, Federica Bovo, Donatella Lombardo, Stefano Amadesi, Tiziana Lazzarotto, Marco Coppi, Gian Maria Rossolini, Simone Ambretti

**Affiliations:** 1Division of Microbiology, IRCCS Azienda Ospedaliero-Universitaria di Bologna, 40138 Bologna, Italy; federica.bovo@aosp.bo.it (F.B.); donatella.lombardo@aosp.bo.it (D.L.); stefano.amadesi@gmail.com (S.A.); tiziana.lazzarotto@unibo.it (T.L.); simone.ambretti@aosp.bo.it (S.A.); 2Clinical Microbiology and Virology Unit, Careggi University Hospital, 50134 Florence, Italy; tommaso.giani@unifi.it (T.G.); marco.coppi@unifi.it (M.C.); gianmaria.rossolini@unifi.it (G.M.R.); 3Department of Experimental and Clinical Medicine, University of Florence, 50100 Florence, Italy; 4Section of Microbiology, Department of Experimental, Diagnostic and Specialty Medicine, University of Bologna, 40100 Bologna, Italy

**Keywords:** novel β-lactams/β-lactamase inhibitors (βL-βLICs), difficult-to-treat (DTR) pathogens, *Enterobacterales*, *P. aeruginosa*, *A. baumannii*, cross-resistance

## Abstract

Multidrug resistance (MDR) represents a serious global threat due to the rapid global spread and limited antimicrobial options for treatment of difficult-to-treat (DTR) infections sustained by MDR pathogens. Recently, novel β-lactams/β-lactamase inhibitor combinations (βL-βLICs) have been developed for the treatment of DTR infections due to MDR Gram-negative pathogens. Although novel βL-βLICs exhibited promising in vitro and in vivo activities against MDR pathogens, emerging resistances to these novel molecules have recently been reported. Resistance to novel βL-βLICs is due to several mechanisms including porin deficiencies, increasing carbapenemase expression and/or enzyme mutations. In this review, we summarized the main mechanisms related to the resistance to ceftazidime/avibactam, meropenem/vaborbactam and imipenem/relebactam in MDR Gram-negative micro-organisms. We focused on antimicrobial activities and resistance traits with particular regard to molecular mechanisms related to resistance to novel βL-βLICs. Lastly, we described and discussed the main detection methods for antimicrobial susceptibility testing of such molecules. With increasing reports of resistance to novel βL-βLICs, continuous attention should be maintained on the monitoring of the phenotypic traits of MDR pathogens, into the characterization of related mechanisms, and on the emergence of cross-resistance to these novel antimicrobials.

## 1. Introduction

Bacterial infections caused by multidrug-resistant (MDR) Gram-negative pathogens have become a major worldwide public health problem during the last two decades [1] due to inadequate therapeutic options that led to increased morbidity, mortality and higher healthcare costs [2]. Against MDR pathogens, carbapenems have been considered the last resort drug for a long time. Carbapenem, and in general β-lactams, act by inhibiting cell wall biosynthesis and are the most used class of antimicrobial agents in the clinic armamentarium for infectious diseases [1]. Carbapenem-resistant *Enterobacterales* (CRE) are classified as a highly critical group of MDR organisms according to the World Health Organization (WHO) antimicrobial resistance report [3]. *Enterobacteriales* species such as *Klebsiella* spp., *Escherichia coli* and *Enterobacter* spp. are a common cause of both community and healthcare-associated infections, and carbapenems are one of the last resources for treatment of extended-spectrum β-lactamase (ESβL) and AmpC producers. For these reasons, the emergence of CRE represents a relevant limitation of therapeutic approaches for the treatment of severe infections in critically ill patients. Carbapenem resistance in *Enterobacterales* is frequently determined by the production of enzymes, so called carbapenemase [4].

Carbapenemase are divided into two different groups on the basis of residues in the active sites: (*i*) serine carbapenemase and (*ii*) Metallo-β-Lactamases (MBL). Following the Ambler classification system, β-lactamases conferring resistance to carbapenem belong to: Class A (mostly KPC), Class B (MBL mostly VIM, NDM and IMP), and Class D (OXA carbapenemase). Although carbapenemase-producing *Enterobacterales* (CPE) prevalence is increasing globally, epidemiology of carbapenemase typically shows wide regional heterogeneity [2]. Based on limited therapeutic options, various previously used drugs, such as fosfomycin and polymyxins, have been renewed for treatment of infections due to MDR pathogens [5]. Simultaneously, development and evaluation of novel combination regimens (e.g., carbapenems and tigecycline) have also been proposed.

Carbapenem-resistant *Pseudomonas aeruginosa* (CR-Pa) and *Acinetobacter baumannii* (CR-Ab) are a leading cause of hospital-acquired infections which are frequently associated with high mortality and morbidity, especially among critically ill patients [1]. Against these MDR pathogens, few antimicrobial molecules exhibit in vitro activity, thus reducing antimicrobial therapy options. Recently, the WHO indicated a priority MDR pathogens list for which new antibiotics are urgently needed by guiding research, and promoted the development of new antibiotics [3]. In this context, different β-lactam–β-lactamase inhibitor combinations (βL-βLICs) were recently developed and approved for the treatment of infections due to MDR micro-organisms [6]. These novel antimicrobial molecules are reported to be active against different MDR pathogens, including class A and D producing *Enterobacterales*, CR-Pa and CR-Ab. Ceftazidime–avibactam (CAZ-AVI) is the first member of this new generation of βL-βLICs. Avibactam is a non–β-lactam β-lactamase inhibitor that restores in vitro activity of a third-generation cephalosporin, ceftazidime, against Ambler class A, C, and some of class D carbapenemase. Subsequently, two novel βL-βLICs, meropenem/vaborbactam (MER-VAB) and imipenem/relebactam (IMI-REL), were registered and approved for treatment of infections due to Gram-negative MDR bacteria with limited treatment options. Both MER-VAB and IMI-REL are based on a combination of carbapenem with novel β-lactamase inhibitors without beta-lactam motif which are able to restore the activity of carbapenem against MDR microorganisms producing class A carbapenemase, while not against MBL producers.

Clinical data describing the efficacy of therapeutic regimens based on novel βL-βLICs for treating infections due to MDR pathogens are promising. However, it should also be stressed that, for these new drugs, different types of resistance mechanisms have already been described and the rapid emergence of resistance to these agents highlights the need for susceptibility in vitro testing, surveillance and application of antimicrobial stewardship strategies.

The aim of this review is to describe the mechanisms that form the basis of resistance to CAZ-AVI, MER-VAB and IMI-REL in Gram-negative MDR pathogens and the microbiological methods to correctly define in vitro susceptibility tests.

## 2. Antimicrobial Agents

### 2.1. Ceftazidime-Avibactam

CAZ-AVI was the first βL-βLICs to be released and was approved for the treatment of complicated intra-abdominal infections (cIAIs) and complicated urinary tract infections (cUTIs) in 2015, and subsequently for the treatment of hospital-acquired and ventilator-associated bacterial pneumonia (HABP/VABP) in 2018 [7]. CAZ-AVI is a novel association of ceftazidime, a third-generation cephalosporin with avibactam, a new reversible (non-suicidal) b-lactamase inhibitor belonging to the diazabicyclo octane class (DBOs). Avibactam (AVI) forms a covalent bond with the serine of the active center of the β-lactamase; however, unlike clavulanic acid and tazobactam, the molecule is not hydrolyzed, but is slowly separated and recovers its original structure. Avibactam is active against class A (ESβLs, KPCs), class C (Amp C, FOX, CMY-2, AAC-1), and class D (OXA-48) β-lactamases, while not active against MBL (e.g., NDM, VIM, IMP) due to the absence of active-site serine residue and against *Acinetobacter* OXA-type carbapenemase [8,9,10,11].

The global surveillance study INFORM (International Network for Optimal Resistance Monitoring) demonstrated that avibactam at concentration of 4 g/L is able to restore ceftazidime activity against 95% of *P. aeruginosa* isolates and 99% of *Enterobacterales* isolates [6]. Previous studies reported that the combination is active against ESβL- and AmpC-producing isolates of *E. coli*, *K. pneumoniae*, *K. oxytoca*, and *P. mirabilis* [12,13]. Additionally, 73% of CRE strains were susceptible to CAZ-AVI [14]. Although CAZ-AVI was recently approved for clinical use, resistance to this novel combination has emerged rapidly in the USA and Europe [15,16]. The rapid emergence of CAZ-AVI-resistant strains represents a serious cause for concern, as highlighted in the Rapid Risk Assessment (RRA) published by the European Centre for Disease Prevention and Control (ECDC) in Stockholm on 12 June 2018 [17]. Resistance to CAZ-AVI in *Enterobacterales* is commonly due to three different mechanisms (Table 1): enzymatic alterations causing inactivation of the antibiotics; modification of the antibiotic target or expressions of an alternative target; and changes in cell permeability or expression of efflux pumps. Modification of β-lactamase hydrolytic properties due to specific mutations within class A carbapenemase is the most common mechanism related to CAZ-AVI-resistance in *Enterobacteriales*, and in combination with the modification of the antibiotic target and the changes in cell permeability can significantly increase the Minimal Inhibitory Concentration (MIC) for CAZ-AVI [18]. Among different enzymatic alterations related to CAZ-AVI-resistance, mutations within the bla_KPC_ gene were the most well-characterized.Amino acid substitutions are commonly observed within the Ω loop of KPC carbapenemase, an important active site of β-lactamases. These mutations mostly occur at amino acid positions 164–179, a conserved structural element that forms the binding cavities with two amino acids (Glu 166 and Asn 170) implied in the acylation and diacylation of substrates by class A β-lactamases. Single amino-acid substitutions in class A β-lactamases at positions 164 and 179 enhance the covalent trapping of the β-lactamases to ceftazidime [19] representing a clinical threat as a potential adaptation to the widespread use of cephalosporins [20]. CAZ-AVI resistance has most frequently been reported in KPC3-producing *K. pneumoniae* belonging to clonal complex (CC)258, a highly successful epidemic clone [21,22]. The KPC variants exhibited higher MICs to CAZ-AVI than other KPC subtypes (MICs for CAZ-AVI ranging from 128 to 256 mg/L), compared to the basal MICs shown by the wild-type KPC variants [23,24,25,26]. Since the clinical approval of CAZ-AVI by the FDA in 2015, various studies have reported the emergence of KPC mutations following antimicrobial therapy [15,24,25,26,27,28]. In 2016, Shields et al. [15] conducted a retrospective study of thirty-seven patients treated with CAZ-AVI for CAZ-AVI-susceptible CRE infections. Authors demonstrated that CAZ-AVI resistance had emerged in three *K. pneumoniae* isolates belonging to the epidemic clone ST258 producing D179Y mutation within the Ω-loop of KPC-3. Of note, different studies demonstrated that D179Y mutation was related to restored susceptibility to meropenem, thus determining a two- to nine-fold reduction in the initial meropenem MICs [24]. In this context, the clinical efficacy of carbapenem-based treatment in patients with infection due to CAZ-AVI-resistant KPC-producing *K. pneumoniae* with reverted phenotype for carbapenem has been proposed [24]. At the same time, reliable and rapid identification of carbapenemase type is essential for the establishment of therapies based on CAZ-AVI treatment. Previous studies reported that false negative detection of KPC production by phenotypic assays (i.e., NG-Test CARBA 5, Neo-Rapid Carb Screen test and DDS assay) occurred in KPC-*K. pneumoniae* strains with subpopulations harboring the D179Y substitution (*bla*_KPC-31_) or alanine-to-threonine substitution at amino-acid 172 (*bla*_KPC-39_) within the Ω-loop of KPC [29,30]. Authors suggested that false negative immunochromatographic tests could suggest a consequence of low binding affinity to mutated KPC enzymes. This point represents a serious cause for concern for the treatment of KPC variants to limit diffusion of CAZ-AVI-resistant strains, avoiding false negative results which may be a cause of therapeutic failure. Mutations in the bla_KPC-3_ gene associated with CAZ-AVI-resistance in patients exposed to prior antimicrobial treatment were also described in other studies [25,31,32]. A list of mutations reported within the *bla*_KPC_ gene are shown in Table 1 [23,24,25,26,33,34,35,36,37]. Regarding strains harboring KPC-2 mutations, Pro169Leu substitution in *K. pneumoniae* and Asn179Asp and Tyr179Asp substitutions in *E. coli* have been reported [35,36,37]. Pneumonia and renal replacement therapy (RRT) are independent risk factors for clinical failure and insurgence of CAZ-AVI-resistant KPC-*K. pneumoniae* isolates [23]. In a single-center study conducted on patients with CRE infections and treated with CAZ-AVI, Shields et al. reported that microbiologic failures occurred in 32% of patients and resistance emerged in 8 out of 77 patients. Interestingly, resistance was observed only in patients with infections due to KPC-3 producers, and mostly due to mutations within KPC (87.5%). A recent study conducted by Coppi et al. demonstrated that CAZ-AVI-resistance was associated with altered outer membrane porins (truncated OmpK35 and an Asp137Thr138 duplication in the L3 loop of OmpK36) and pKpQIL plasmid harboring two copies of the Tn*4401*-KPC-3-encoding transposon [38]. Concurrently, Sun et al. demonstrated that although resistance to CAZ-AVI was due to mutations in the *bla*_KPC_ gene, the increased gene expression and copy number of mutated *bla*_KPC_ genes was associated with the highest MIC for CAZ-AVI (2048 mg/L) [39]. Recently, selection of subpopulations of KPC-producing *K. pneumoniae* resistant to CAZ-AVI has been demonstrated to be associated with suboptimal drug exposure in a critically ill patient with a pneumonia infection [40].

Resistance to CAZ-AVI is also reported in other class A (CTX-M or SHV) and class C (Amp C) β-lactamases. Previous studies demonstrated that CAZ-AVI resistance is associated with at least two amino acid substitutions in ESβL genes, namely Ser130Gly and Leu169Gln substitutions in CTX-M-15, and Pro170Ser and Thr264Ile mutations in CTX-M-14. In addition, single mutations in SHV (Ser130Gly) have been related to CAZ-AVI-resistance in *E. coli* [41,42,43,44].

Resistance to CAZ-AVI has been also reported in *E. coli*, *C. freundii*, *E. cloacae* and *E. aerogenes strains* harboring AmpC mutations. Previous studies demonstrated several amino acids mutations within the AmpC Ω loop [45]. In particular, mutations in AmpC, such as Arg168His in *C. freundii*, Gly176Arg/Asp substitution and a six-residue deletion in the H-10 helix in *E. cloacae,* increase the MICs of CAZ-AVI [40,46]. Structural alterations in the R2 binding site, H-9 and H-10 helices, and Tyr150Cys substitution in *E. coli* also led to CAZ-AVI non-susceptibility [47,48].

**Table 1 antibiotics-11-00628-t001:** Mutations and structural modifications related to the resistance mechanisms for ceftazidime-avibactam (CAZ-AVI) in Gram-negative MDR bacilli.

Ambler Class Classification	ꞵ-Lactamase Reference	Pathogen	Mutations and/or Modifications
A	KPC-3	*Enterobacterales*	V240G: Ala for Val substitution at amino acid position 240 [15] D179Y: Tyr-for-Asp acid substitution at amino acid position 179 within the KPC-3V loop [22,23,24] V240G: Gly for Val substitution at amino acid position 240 [22] A177E: Glu for Ala substitutions at KPC-3 177 positions 177 [24] T243M: Met for Thr substitution at position 243 [33] 165–166 EL: Glu and Leu insertion between positions 165 and 166 [33] V240A: Ala for Val substitution at amino acid position 240 [35] A179T: Thr-for-Ala substitution at amino acid position 179 [49] R164S: Arg-for-Ser substitution at amino acid position 164 [49] S272insKDD: KDD triplet insertion at position 272 [50,51] S272insKDDKDD: KDDKDD triplet insertion at position 272 [50,51] L167delEL: EL residue deletion at position 167 [51] S182insSS: SS amino acid residue duplication at position 182 [51] 269-ProAsnLys-270: 3-amino-acid insertion between positions 269 and 270 [52] 276-Glu-Ala-Val-277: 3-amino-acid insertion between positions 276 and 277 [53] L168insLE: LE amino acid residue duplication at position 168 [54] L168insLELE: two copies of LE amino acid residue duplication at position 168 [54]
	KPC-2	*Enterobacterales*	D179N: Asn for Asp acid substitution at amino acid position 179 [28] D179V: Val for Asp acid substitution at amino acid position 179 [28] D179A: Ala for Asp acid substitution at amino acid position 179 [28] L169P: Pro for Leu substitution at amino acid position 169 [35] D179Y: Tyr for Asp acid substitution at amino acid position 179 [33,55] Δ242-GT-243: GT deletion at positions 242 and 243 [56]
	CTX-M	*Enterobacterales*	D182Y: CTX-M-15 mutation: Asp for Tyr substitution at amino acid position 182 [41]; L169Q and S130G: Gln for Leu substitution at amino acid position 169 and Gly for Ser substitution at amino acid position 130 [42] P170S and T264I: CTX-M-14 mutation: Pro for Ser substitution at amino acid position 170; Thr for Ile substitution at amino acid position 264 [43]
	SHV	*Enterobacterales*	S130G: Ser130Gly: lack of a hydroxyl group at position 130 slows carbamylation of the enzyme by avibactam [44].
	VEB	*K. pneumoniae*	K234R: Arg for Lys acid substitution at amino acid position 234 [57]
		*P. aeruginosa*	
C	AmpC	*P. aeruginosa*	The changes in the V loop are expected to influence both ceftazidime hydrolysis and avibactam inhibition [45]. Mutations in positions such as amino acids 168, 176, 309–314 and 366 lead to non-susceptibility; G168R: Arg168His (and Gly176Arg/Asp) raised CAZ-AVI MICs [41].
		*Enterobacterales*	
		*Enterobacterales*	Structural alterations in the R2 binding site and H-9 and H-10 helices, which are secondary structures surrounding the R2 binding site [47].
			CHE: contains a six-residue deletion in the H-10 helix in close proximity to the active site [46].
			N346Y and Y150S: Asn for Tyr substitution at amino acid position 346 or a Tyr for Ser substitution at amino acid position 150, which results in a steric clash with the sulphate group of avibactam, thus influencing the binding affinity of the inhibitor [48].
			Y150 C: CMY-6: Tyr for Cys substitution at amino acid position 150 [48]
			N346I: CMY-10: Asn for Ile substitution in helix H-11 position 346 [42]
D	OXA-2	*P. aeruginosa*	OXA-539: duplication of the key residue Asp149 [58]
	OXA-48-family	*Enterobacterales*	P68A and Y211S: Ala for Pro substitution at amino acid position 68 and Ser for Tyr substitution at amino acid position 211 coexist [59].
	OXA-51	*A. baumannii*	[46]

Table 1 shows class D (OXA) β-lactamase mutations in *P. aeruginosa* [58], *E. coli* [59], and *A. baumannii* [60] associated with CAZ-AVI resistance.

Another important mechanism associated with CAZ-AVI-resistance is membrane permeability due to decreased expression and/or mutations in porin genes and overexpression of efflux systems. Previous studies demonstrated that mutations of OmpK35 and OmpK36 porins significantly increased the MIC for CAZ-AVI in *K. pneumoniae* [39,61,62]. In particular, CAZ-AVI-resistance has been associated to variance in OmpK36, caused by a duplication or insertion of two amino acids (Gl134-D135) in the L3 loop, insertional inactivation IS5 that decreases the expression of OmpK36, or lack of OmpK35, which has an early frameshift causing a premature stop codon [30,61,62]. These porin mutations often require the presence of other mechanisms to increase the MIC significantly, such as OmpK36 and ESβL in *K. pneumoniae* [63], or OprD loss and elevated AmpC expression in *P. aeruginosa* [64].

Alterations in efflux pumps have been demonstrated to be related to CAZ-AVI resistance in *K. pneumoniae* and *P. aeruginosa* [58]. Although efflux pumps do not seem to solely have a role in CAZ-AVI resistance in *Enterobacterales*, Winkler et al. showed that efflux pump inhibitors CCCP and PaβN contributed to resistance to CAZ-AVI in *P. aeruginosa* [65,66]. Concurrently, Chaloub et al. demonstrated that increased MIC of CAZ-AVI in AmpC-producing *P. aeruginosa* was associated with increased activity of avibactam efflux transporters due to an overexpressing MexAB-OprM system associated with increased AmpC expression, while excluding the role of OprD porin [67].

Target protein mutations seem to be related to the increasing MIC for CAZ-AVI in *E. coli*, *P. aeruginosa*, *H. influenzae*, *S. aureus* and *S. pneumoniae*. Previous studies demonstrated that avibactam binds covalently to various PBPs; as PBP2 of *E. coli*, *H. influenzae* and *S. aureus*; PBP2 and PBP3 of *P. aeruginosa*; and PBP3 of *S. pneumoniae*. however, ceftazidime instead mainly binds to PBP3 [68]. In this context, Alm et al. showed that four-amino-acid insertion (Thr-Ile-Pro-Tyr) into PBP3 of *E. coli* strains appears to play a potential role in CAZ-AVI resistance [69]. This insertion was identified in multiple MLST lineages of *E. coli* mostly producing NDM carbapenemase. However, PBP3 insertions have not yet been reported to be related to resistance to ceftazidime-avibactam, and structural analysis suggests that these changes will impact the accessibility of β-lactams to the transpeptidase pocket of PBP3.

### 2.2. Meropenem-Vaborbactam

MER-VAB is a novel βL-βLICs approved in 2017 by the FDA, and by EMA in 2018, for the treatment of cUTIs including AP, cIAI, HAP and VAP [70,71,72]. MER-VAB represents a valid alternative for the treatment of many infections due to CRE [73]. Vaborbactam is a boronic acid, non-β-lactam β-lactamase inhibitor [74], which exhibited potent activity against the KPC enzyme [74]. It is effective in inhibiting class A and C β-lactamases, in particular the KPC enzyme [75], while CPE producing class D or class B carbapenemase are usually resistant to MER-VAB [76]. In vitro studies demonstrated that vaborbactam at concentration of 8 mg/L restores meropenem activity against carbapenem-resistant strains producing KPC [77]. A large in vitro study conducted by Hackel et al. in 2018 showed that vaborbactam restored meropenem activity in 99% of KPC-producing *Enterobacteriales* isolates [78], while a study conducted between 2013 and 2014 in New York City revealed that 99% of KPC-producing *K. pneumoniae* (KPC-Kp) isolates were susceptible to MER-VAB [79]. Sabet et al. evaluated the in vivo activity of meropenem alone and in combination with vaborbactam in mouse thigh and lung infection models due to KPC-producing carbapenem-resistant strains (i.e., *K. pneumoniae, E. coli*, and *E. cloacae*). Authors demonstrated that meropenem alone did not produce bacterial killing, while the addition of vaborbactam to meropenem exerted higher bactericidal activity against strains with MER-VAB MIC up to 8 mg/L [80]. MER-VAB safety and efficacy was evaluated in patients for the treatment of cUTIs and pyelonephritis. In particular, the TANGO-I study (a multicenter randomized double-blind non-inferiority study conducted from 2014 to 2016) concluded that MER-VAB was statistically superior to piperacillin/tazobactam (PIP/TAZ) for the treatment of cUTIs (98.4% and 94%, respectively), while the safety level was similar to PIP/TAZ. TANGO-II, a multicenter randomized open-label study conducted between 2014 and 2017, compared the efficacy and safety of MER-VAB with best available therapy for the treatment of severe infections due to CRE. Results showed that MER-VAB significantly improved clinical cure and mortality rates, demonstrating a lower level of nephrotoxicity than best available therapy (BAT) (11.1% vs. 24%) [81].

Antibiotic resistance may occur throughout different inherent structural or functional characteristics of bacterial species [82], such as enzymatic degradation, modifications of antibiotic target site, activation of efflux pumps and alteration or interference of the antibiotic intake [81].

To date, the main mechanism associated with MER-VAB resistance in KPC-producing *Enterobacterales* is impaired permeability due to porin mutations associated with overexpression of β-lactamase and increases in efflux pump production [83,84].

A recent study conducted by Dulyayangkul et al. revealed that *kvrA* inactivation, and subsequently OmpK35/36 porins downregulation, can affect the antimicrobial susceptibility to MER-VAB in KPC-3-producing *K. pneumoniae* isolates [85]. Although loss of expression of the OmpK35, OmpK36 and/or OmpK37 porins has been associated with MER-VAB-resistance in KPC-Kp strains, the role of different porins has been recently demonstrated [63,77]. In particular, the OmpK36 porin, which has a smaller channel than OmpK35, appears to be more significant in the influx of vaborbactam across the outer membrane [86]. Lapuebla et al. found that the activity of vaborbactam was reduced in KPC-Kp isolates with a decreased expression of OmpK36 in comparison to the same KPC-producing isolates with functional porins [79]. A subsequent study conducted by Lomovskaya et al. on KPC-3-producing *K. pneumoniae*, showed that mutant prevention concentration (MPC) of MER-VAB-resistance increased 64-fold and 4-fold, respectively, when *ompK36* and *ompK35* genes were inactivated alone.

Among the different mutations occurring in KPC-Kp isolates, several studies demonstrated that the most frequent mutation resulting in a non-functional OmpK35 porin is deletion of A at nucleotide 86 that caused a frameshift from amino acid 29 (FS_aa29). In addition, the most frequent mutations identified in OmpK36 are glycine (G) and aspartic acid (D) insertion at position 134–135 [77,87,88,89]. These amino acid duplications have been identified in the conserved L3 loop of *ompK36,* which serves as ion selection of the pore. This domain forms a constriction zone within the channel that contributes to the permeability properties of the porins and forms a bottleneck for carbapenems [72,77,88].

In a preliminary prospective observational study, Shields et al. observed that, in patients with CRE infections, clinical success and survival rates were observed in 65% (13/20) and 90% (18/20), respectively, of patients treated with MER-VAB. Of note, microbiological failure occurred in a patient harboring an ST258 strain of KPC-31-producing *K. pneumoniae* after 12 days of treatment (MER-VAB MIC 0.12 mg/L to 8 mg/L), and whole genome sequencing identified an IS5 insertion in the *ompK36* promoter, confirming the important role of this protein in reducing MER-VAB susceptibility [88].

As discussed above, although resistance to MER-VAB has been associated with decreased expression of *ompK3*5 and *ompK36* and concomitantly increased expression of *bla_KPC_*, MIC seems to be unaffected by an increase in expression of the *bla_KPC_* gene and efflux pump (*acrB*), or decreased expression of *ompK35* alone [72]. Sun et al. demonstrated that in vitro mutant selection of KPC-Kp strains with increased MIC to MER-VAB exhibited *ompK36* inactivation or partially functional *ompK36* associated with increased *bla*_KPC_ gene copy number [89]. In this context, three main mechanisms have been identified to determine the increase in copies of the *bla*_KPC_ gene: (i) intracellular transposition of Tn4401 that carries *bla*_KPC_ from a large low copy number plasmid to a much smaller high copy number plasmid; (ii) increase in the number of copies of *bla*_KPC_ per plasmid, or increase in the number of KPC-carrying plasmids per cell by internal rearrangements of a KPC-carrying plasmid; and (iii) insertional inactivation of the *repA2* gene, which controls plasmid replication [89].

Among *Enterobacterales* and other Gram-negative bacteria efflux pump systems, in particular AcrAB-TolC, are common resistance mechanisms against multiple antibiotic classes [90]. Lomovskaya et al. assessed the in vitro activity of MER-VAB in *K. pneumoniae* harboring different porin protein mutations and multidrug resistance efflux pumps. Authors demonstrated that downregulation of ompK35 and overexpression of acrAB, due to mutation in the *ramR* gene, did not affect the activity of MER-VAB, while overexpression of acrAB in association with inactivated ompK35 and ompK36 porins increased the MIC of MER-VAB [73]. The effect of a combination of multiple resistance mechanisms against MER-VAB in *K. pneumoniae* isolates has been illustrated in a study conducted by Zhou et al. in 2018 showing that MIC of MER-VAB was not affected by diminished OmpK35 or increased expression of *bla*_KPC_ or *acrB* alone, while strains showing a complete inactivation of porins in combination with increased expression of *bla*_KPC_ and *acrB* genes were associated with the highest MIC for MER-VAB [77].

### 2.3. Imipenem-Relebactam

IMI-REL is a recent βL-βLICs approved by the FDA in 2019 [91], and by the EMA in 2020 [92], for treatment of cUTI, cIAI, HAP and VAP with limited or no alternative therapeutic options caused by multi-resistant Gram-negative bacteria [93]. Relebactam (formerly described as MK-7655) is a non-β-lactam bicyclic diazabicyclooctane (DBO) β-lactamase inhibitor. It is structurally similar to avibactam, except for the addition of a piperidine ring conceived to prevent the efflux of this molecule from bacterial cells [94,95].

The IMI-REL combination has shown effective in vitro activity against ß-lactamases belonging to Ambler’s class A (such as KPC, TEM, SHV and CTX-M) and class C (AmpC, CMY). On the other hand, relebactam is not active against class B MBL (NDM, VIM and IMP) and has limited activity against class D (OXA-48-like) carbapenemase [96,97,98].

Clinical data studies demonstrated that IMI-REL is associated with favorable clinical response and safety in patients for treatment of imipenem-nonsusceptible infections [99]. In particular, the RESTORE IMI-1 clinical trial reported the non-inferiority and well-tolerance of IMI-REL compared to imipenem plus colistin for infections due to imipenem-resistant pathogens, while the RESTORE IMI-2 study reported the efficacy and safety of IMI-REL in treating hospital-acquired/ventilator-associated bacterial pneumonia (HABP/VABP) in comparison to piperacillin-tazobactam [100,101].

At a fixed concentration of 4 mg/L, relebactam is able to restore imipenem activity against 92.7% of KPC-producing *Enterobacteriales* [102]. Currently, EUCAST established the clinical breakpoint to IMI-REL at 2 mg/l for resistant isolates [103].

To date, a limited number of carbapenemase-producing Enterobacterales resistant to IMI-REL have been described. Among different mechanisms, class B and D carbapenemases are the main cause of IMI-REL resistance in CRE. As discussed above, strains producing these carbapenemases are often resistant to IMI-REL [104,105]. Several studies demonstrated that IMI-REL resistance can also be due to different mechanisms which include: carbapenemase mutation, carbapenemase over-expression, penicillin binding proteins (PBPs) mutation or under-expression, increased efflux and decreased permeability.

A recent study [106] conducted on CRE demonstrated that resistance to IMI-REL is associated with KPC-3 and SME-1 production in *Serratia marcescens*. Authors identified six isolates harboring KPC-3 (MIC of 2 mg/L) and one harboring SME-1, which lead to the highest IMI-REL MIC in the study (4 mg/L).

Previous studies demonstrated that IMI-REL resistance was associated with mutations resulting in a non-functional OmpK35 and OmpK36 porins in KPC-Kp strains. In particular, Lapuebla et al. demonstrated that loss of OmpK36 in KPC-Kp was associated with IMI-REL resistance (IMI-REL MIC 8 mg/L) [107]. Balabanian et al. reported that major disruptions in both OmpK35 and OmpK36 porins correlated to reduced activity of IMI-REL (MICs 2/4, 8/4, and 512/4 mg/L) in three KPC-Kp strains also harboring SHV variants (SHV-11 and SHV-12) and TEM-1 [108]. Of note, the strain exhibiting high MIC for IMI-REL showed *bla*_KPC_ over-expression and acrB efflux pump downregulation.

A subsequent study conducted by Galani et al. showed that although the KPC enzyme is inhibited by relebactam, resistance to IMI-REL can emerge as a consequence of chromosomal factors such as OmpK35 disruption and OmpK36 mutation [102]. Authors tested IMI-REL against KPC-Kp and found that six isolates (2%) exhibited high IMI-REL MICs (4 mg/L). Among these isolates, five harbored blaKPC-2 and one *bla*_KPC-23_. Wild-type OmpK35 was detected in a single isolate, while the others had a truncated protein. Regarding OmpK36, four isolates harbored a wild-type protein. A single isolate had an OmpK36 porin with a GD134-135 insertion correlated to high carbapenem resistance. Additionally, another isolate exhibited OmpK36 with an A323P amino acid substitution [102].

In recent studies, we described the dynamic evolution of a KPC-Kp strain resistant to IMI-REL in patients following CAZ-AVI-based treatment [109,110]. Interestingly, resistance to IMI-REL resistance evolved with the evolution of different *bla*_KPC_-mutated subpopulations associated to transposition events of the Tn*4401* harboring region. In these cases, resistance to IMI-REL was due to an increased copy number of *bla*_KPC_ in a KPC-Kp strain harboring disrupted OmpK35 and GD134-135 inserted OmpK36 porin. Similar findings were recently described in a hematological patient with bloodstream infections due to KPC-Kp cross-resistant to IMI-REL and MER-VAB and was successfully treated with CAZ-AVI in combination with gentamicin [111].

A previous study demonstrated that AmpC overexpression in combination with porins loss has been related to IMI-REL resistance [101]. Authors reported a resistant (IMI-REL MIC 4 mg/L) carbapenemase-negative *K. aerogenes* isolate harboring disrupted OmpK35 and OmpK36 porins and exhibiting AmpC overexpression.

Regarding carbapenemase-producing *Pseudomonas aeruginosa*, two distinct studies reported the emergence of IMI-REL-resistant strains producing *bla*_GES-5_ [112,113]. A large in vitro study conducted by Fraile-Ribot et al. demonstrated that IMI-REL exhibited potent activity against *P. aeruginosa* mutants with AmpC hyperproduction (such as AmpD and PBP4 mutants), OprD inactivation, and/or efflux pump (MexAB-OprM, MexXY, and MexCD-OprJ) overexpression and that IMI-REL-resistance was associated to cross-resistance to ceftolozane-tazobactam and CAZ-AVI. On the other hand, isolates producing carbapenemases such as VIM, IMP and GES-5, proved resistant to IMI-REL (MIC > 8 mg/L). At the same time, authors reported that a carbapenemase-nonproducing—*P. aeruginosa* overexpressing MexXY system (due to mexZ inactivation) and ampC (due to PBP4 mutation) in association with unique mutations in PBP 2 (A269V) and PBP 3 (N242S) exhibited increased MIC for IMI-REL (MIC of 8 mg/L) [113].

Mushtaq et al. showed that although relebactam reversed imipenem resistance against KPC-producing *P. aeruginosa*, as observed in *Enterobacterales*, moderated reduction of activity has been observed for *P. aeruginosa* producing ESβL enzymes (VEB, PER, GES and SHV). Moreover, isolates harboring GES-5 carbapenemase exhibited high IMI-REL MICs (ranging from 32 to 128 mg/L) remaining far beyond the clinical range [114]. Contrastingly, in carbapenemase-negative *P. aeruginosa* isolates, resistance to IMI-REL is mainly caused by OprD porin depletion [115].

Lapuebla et al. demonstrated that OprD porin downregulation is associated with reduced susceptibility to IMI-REL in *P. aeruginosa*. Authors evaluated the effect of IMI-REL against a collection of *P. aeruginosa* isolates with reduced oprD expression and varying AmpC expression, concluding that relebactam could not effectively restore imipenem activity (IMI-REL MICs ranging from 0.25 to 8 mg/L). At the same time, AmpC expression did not seem to affect IMI-REL MICs [107].

A recently published paper [116] demonstrated that *P. aeruginosa* can develop resistance to IMI-REL through acquisition of carbapenemase-encoding genes. Authors compared the genomes of a wide number of *P. aeruginosa* isolates based on their IMI-REL susceptibility. They observed that resistant carbapenemase-positive isolates harbored the class B carbapenemase VIM-4, which is usually contained in mobile genetic elements (MGEs) that facilitate its spread. Additionally, the study confirmed that alterations in OprD porin lead to IMI-REL-resistance in carbapenemase-negative isolates.

## 3. Susceptibility Test for Novel β-Lactams/β-Lactamase Inhibitor

Antimicrobial susceptibility testing (AST) is a crucial activity for the clinical diagnostic laboratory of microbiology. AST can be performed via different methods including broth microdilution, agar dilution, disk diffusion, gradient strip diffusion (using different support: paper or plastic) or automated systems. The results of AST (minimum inhibitory concentrations or growth inhibition zone diameters around disks) are translated into susceptibility categories according to the clinical breakpoints defined by various committees (e.g., CLSI or EUCAST) and are used to predict clinical efficacy of the tested antibiotics. For these reasons, accurate AST results have crucial importance. Problems of low accuracy of AST using different methods versus reference methodologies (e.g., broth microdilution (BMD) or agar dilution for fosfomycin) have previously been reported for different clinically important molecules such as colistin, tigecycline, gentamycin, fosfomycin or vancomycin. Unfortunately, reference methods are not always used for routine AST because of the additional workload required compared to automated systems or other manual methods. Moreover, for the novel β-lactams/β-lactamase inhibitors not a different tests are not always available because of their recent introduction in the clinical practice. Thus, it is important to know the performance of the different tests (easier to adopt in the routine workflow) used in the clinical microbiology laboratory vs. reference methodologies to evaluate susceptibility to the new antibiotic molecules.

### 3.1. Ceftazidime/Avibactam Susceptibility Testing

The European Committee on Antimicrobial Testing (EUCAST) approved CAZ-AVI species-related breakpoints for *Enterobacterales* and *P. aeruginosa* (S: ≤8/4 mg/L and >8/4 mg/L corresponding to zone diameters of S: ≥13 mm and R: <13 mm using 10/4 µg disk content of CAZ-AVI) [103,117]. The CLSI committee also set breakpoints for *Enterobacterales* and *P. aeruginosa* (S: ≤8/4 mg/L and ≥16/4 mg/L corresponding to zone diameters of S: ≥21 mm and R: ≤20 mm using 30/20 µg a disk content of CAZ-AVI).

Broth microdilution technique determined as recommended by ISO 20776-1 guideline and using a fixed concentration of 4 mg/L of the avibactam inhibitor, is considered the gold standard method for CAZ-AVI AST [118]. EUCAST suggests the use of the *K. pneumoniae* ATCC700603 (an SHV-18 ESβL producer) as control of the inhibitor component.

Several commercial CE-IVD and/or FDA approved testing devices for CAZ-AVI AST are on the market and can be used in diagnostic laboratories. Broth microdilution panels (Sensititre by Thermofisher Scientific and Merlin Diagnostika), gradient diffusion tests (MIC-test-strips by Liofilchem and E-test by bioMèrieux), disk diffusion tests (form several companies) and automated AST panels (Microscan, Vitek and Phoenix platforms) have been developed.

Studies that evaluated the performance of gradient strip diffusion vs. reference method for CAZ-AVI AST showed a good correlation between the two methods with a Category Agreement (CA) and Essential Agreement (EA) ranging from 77–99% and 85–100%, respectively (with lower performances registered for MIC test strip) (Table 2). Furthermore, the low number of Major errors (ME) and Very Major errors (VME) reported when carbapenemase producing *Enterobacterales* or *P. aeruginosa* were also tested, suggests that gradient strip tests are suitable devices for routine tests of CAZ-AVI for *P. aeruginosa* and *Enterobacterales*.

Overall disk diffusion (DD) vs. reference BMD, both using CLSI or EUCAST disk content, showed lower performance than the gradient strip test, with the tendency of overestimating resistance (higher number of ME). For these reasons some of the authors concluded that the DD results for CAZ-AVI, especially in case of carbapenemase producers, should be interpreted cautiously. Automated systems for CAZ-AVI AST showed good correlation with BMD with CA and EA values ranging from 83–100% and 87–100% even if the evaluations present in the literature are few.

### 3.2. Meropenem/Vaborbactam Susceptibility Testing

The EUCAST committee approved MER-VAB species-related breakpoints for *Enterobacterales* and *P. aeruginosa* (S: ≤8/8 mg/L and >8/8 mg/L corresponding to zone diameters of S: ≥18 mm and R: <14 mm using 20/10 µg disk content of MER-VAB) [103,117]. The CLSI committee set breakpoints for *Enterobacterales* only (S: ≤4/8 mg/L and ≥16/8 mg/L corresponding to zone diameters of S: ≥21 mm and R: ≤20 mm using 30/20 µg a disk content of CAZ-AVI).

Broth microdilution technique, determined as recommended by the ISO 20776-1 guideline and using a fixed concentration of 8 mg/L of the vaborbactam inhibitor (that should be solved in a solution of DMSO 90% plus 10% water), is considered as the gold standard method for MER-VAB AST [118]. EUCAST suggests the use of *K. pneumoniae* ATCC BAA-2814 (a KPC-3 carbapenemase producer) as control of vaborbactam inhibitor.

Even if several commercial CE-IVD and/or FDA approved tests for MER-VAB AST are on the market, including broth microdilution panels (Sensititre by Thermofisher Scientific), gradient diffusion tests (MIC-test-strips by Liofilchem and E-test by bioMèrieux), disk diffusion tests (form several companies) and automated AST panels (Microscan, Vitek and Phoenix platforms), their development is very recent. Very few evaluations of the different AST methods are present in the literature. Gradient tests seem to be a valid alternative for MER-VAB AST, with the E-test showing better performance when compared to gradient tests by Liofilchem (that demonstrated a tendency to overestimate MIC) (Table 2). However, the use of the E-test for *Proteus mirabilis* should be discouraged due to unacceptable analytical performance (very low EA: 37%) [132].

### 3.3. Imipenem/Relebactam Susceptibility Testing

The EUCAST committee approved IMI-REL species-related breakpoints for *Enterobacterales* (except *Morganellaceae*), *P. aeruginosa* (S: ≤2/4 mg/L and >2/4 mg/L corresponding to zone diameters of S: ≥22 mm and R: <22 mm using 10/25 µg disk content of IMI-REL) and *A. baumannii* (S: ≤2/4 mg/L and >2/4 mg/L corresponding to zone diameters of S: ≥24 mm and R: <24 mm using 10/25 µg disk content of IMI-REL) [103,117]. The CLSI committee set breakpoints for *Enterobacterales* (S: ≤1/4 mg/L and ≥4/4 mg/L corresponding to zone diameters of S: ≥25 mm and R: ≤20 mm using 10/25 µg disk content of IMI-REL) and *P. aeruginosa* (S: ≤2/4 mg/L and ≥8/4 mg/L corresponding to zone diameters of S: ≥23 mm and R: ≤19 mm using 10/25 µg a disk content of IMI-REL).

IMI-REL is the most recently introduced to the market among the molecules discussed in this paper. Few devices are available for IMI-REL AST including disk diffusion tests (Hardy Diagnostic), gradient tests (MIC-test-strips by Liofilchem and E-test by bioMèrieux) and broth microdilution panels (Sensititre by Thermofisher Scientific) while automated panel systems are under development. Authors that tested different methods concluded that the E-test yielded results comparable to BMD for IMI-REL (Table 2).

## 4. Conclusions

In this review, we summarize the mechanisms at the basis of the resistance to novel βL-βLICs recently developed for the treatment of infections due to multidrug resistant Gram-negative bacteria. In this context, the rapid dissemination of MDR Gram- negative micro-organisms represents a serious threat to global health due to their difficult-to-treat (DTR) resistance phenotypes. Against DTR micro-organisms, novel βL-βLICs represent the key to overcoming this emerging scenario. However, the recent reporting of MDR strains resistant to βL-βLICs posed the necessity of deeper understanding of the mechanisms related to the reduced susceptibility to these antimicrobials. In the context of the rapid emergence of such types of resistance, a continuous monitoring of the susceptibility to these novel combinations is necessary to trace and limit the diffusion of these antimicrobial resistance determinants. At the same time, we believe that phenotypic and genotypic characterizations of novel antimicrobial resistance traits represent a fundamental tool for enlarging knowledge about these novel antimicrobials, to establish a real picture of the epidemiology of related resistance mechanisms, and to define the events at the basis of the acquisition of novel resistance mechanisms.

Finally, particular attention needs to be paid to the emergence of MDR strains resistant to different novel βL-βLICs. Indeed, emerging cross-resistance poses further limitations for the use of novel antimicrobials and has important implications for the correct usage of such molecules to prevent the development and spread of emerging pan-resistant strains.

## Figures and Tables

**Table 2 antibiotics-11-00628-t002:** Evaluation of different commercial methods vs. reference technique for AST of ceftazidime/avibactam (CAZ-AVI), meropenem/vaborbactam (MER-VAB) and imipenem/relebactam (IMI-REL).

Antibiotic Molecule	Evaluated Method	Tested Species (Number; Principal Phenotype If Present)	Evaluation Result ^a^	References
CAZ-AVI	Etest (bioMérieux)	*Enterobacterales* (*n* = 140; 28 ESβL, 23 CP ^b^); *P. aeruginosa* (*n* = 60; 18 CP ^b^)	EA = 99% for *Enterobacterales* EA = 98% for *P. aeruginosa* CA = 100% for *Enterobacterales* (ME = 0; VME = 0) CA = 98% for *P. aeruginosa* (ME = 0; VME = 1)	[119]
CAZ-AVI	Etest (bioMérieux)	*Enterobacterales* (*n* = 74; 74 CR ^c^)	EA =89% CA = 97% (ME = 0; VME = 2)	[120]
CAZ-AVI	Etest (bioMérieux)	*Enterobacterales* (*n* = 194); *P. aeruginosa* (*n* = 77)	EA = 96% for *Enterobacterales* EA = 95% for *P. aeruginosa* CA = 99% for *Enterobacterales* (ME = 1; VME = 0) CA = 96% for *P. aeruginosa* (ME = 3; VME = 0)	[121]
CAZ-AVI	Etest (bioMérieux)	Gram-negatives (*n* = 102; 69 CR ^c^)	EA =77% CA = 95% (ME = 6.3%; VME = 0)	[122]
CAZ-AVI	Etest (bioMérieux)	*Enterobacterales* (*n* = 228); *P. aeruginosa* (*n* = 74)	EA = 97% for *Enterobacterales* EA = 99% for *P. aeruginosa* CA = 100% for *Enterobacterales* CA = 99% for *P. aeruginosa* (ME = 1; VME = 0)	[123]
CAZ-AVI	Etest (bioMérieux)	*N* = 192 *P. aeruginosa*	EA = 95% CA = 94% (ME = 6; VME = 5)	[124]
CAZ-AVI	Etest (bioMérieux)	*N* = 458 *Enterobacterales*	EA = 95% CA = 99% (ME = 1; VME = 1)	[125]
CAZ-AVI	Etest (bioMérieux)	*N* = 200 *P. aeruginosa*	EA = 94% CA = 95% (ME = 5; VME = 5)	[126]
CAZ-AVI	Etest (bioMérieux)	*N* = 187 CR ^c^ *K. pneumoniae*; *n* = 28 CR ^c^ *E. coli*; *n* = 81 CR ^c^ *P. aeruginosa*	EA = 95% for *K. pneumoniae* EA = 96% for *E. coli* EA = 86.4% for *P. aeruginosa*	[127]
CAZ-AVI	MIC test Strip (Liofilchem)	*N* = 192 *P. aeruginosa*	EA = 89% CA = 86 (ME = 25; VME = 2)	[124]
CAZ-AVI	MIC test Strip (Liofilchem)	*N* = 200 *P. aeruginosa*	EA = 95% CA = 93% (ME = 1; VME = 8)	[126]
CAZ-AVI	DD ^d^ (30/20µg disk content, Oxoid)	*Enterobacterales* (*n* = 228); *P. aeruginosa* (*n* = 74)	CA = 100% for *Enterobacterales* CA = 96% for *P. aeruginosa* (ME = 0; VME = 3)	[123]
CAZ-AVI	DD ^d^ (10/4µg disk content, Mast Group)	Gram-negatives (*n* = 102; 69 CR)	CA = 87% (ME = 16%; VME = 0)	[122]
CAZ-AVI	DD ^d^ (30/20µg disk content, Hardy Diagnostic)	Gram-negatives (*n* = 102; 69 CR)	CA = 80% (ME = 25%; VME = 0)	[122]
CAZ-AVI	DD ^d^ (30/20µg disk content, Oxoid)	*Enterobacterales* (*n* = 194); *P. aeruginosa* (*n* = 77)	CA = 98% for *Enterobacterales* (ME = 1; VME = 2) 93% for *P. aeruginosa* (ME = 4; VME = 1)	[121]
CAZ-AVI	DD ^d^ (30/20 µg disk content, Hardy Diagnostic)	*Enterobacterales* (*n* = 74; 74 CR)	CA = 76% (ME = 18; VME = 0)	[120]
CAZ-AVI	DD ^d^ (30/20 µg disk content, Mast Group)	*n* = 500 *Enterobacterales*; *n* = 349 *P. aeruginosa*	ME = 0; VME = 0.4% for *Enterobacterales* ME = 2.9%; VME = 2.3% for *P. aeruginosa*	[128]
CAZ-AVI	DD ^d^ (10/4 µg disk content, Mast Group)	*n* = 192 *P. aeruginosa*	CA = 80% (ME = 38; VME = 1)	[124]
CAZ-AVI	DD ^d^ (10/4 µg disk content, Oxoid)	*n* = 192 *P. aeruginosa*	CA = 88% (ME = 22; VME = 1)	[124]
CAZ-AVI	DD ^d^ (30/20 µg disk content, Oxoid)	*N* = 458 *Enterobacterales*	CA = 99% (ME = 0; VME = 1)	[125]
CAZ-AVI	DD ^d^ (10/4 µg disk content, Thermo Fisher)	*n* = 200 *P. aeruginosa*	CA = 85% (ME = 14; VME = 0)	[126]
CAZ-AVI	DD ^d^ (10/4 µg disk content, Biorad)	*n* = 200 *P. aeruginosa*	CA = 85% (ME = 18; VME = 0)	[126]
CAZ-AVI	Vitek 2 system (bioMérieux, AST-XN12 card)	*n* = 200 *P. aeruginosa*	EA = 89% CA = 83% (ME = 3; VME = 30)	[126]
CAZ-AVI	Vitek 2 system (bioMérieux, AST-GN card)	*n* = 1073 (*n* = 866 *Enterobacterales*; *n* = 207 *P. aeruginosa*)	EA = 94% for *Enterobacterales* EA = 96% for *P. aeruginosa* CA = 99% for *Enterobacterales* (ME = 7; VME = 0) CA = 97% for *P. aeruginosa* (ME = 7; VME = 0)	[129]
CAZ-AVI	BD Phoenix system (BD Diagnostic Systems, NMIC-500 panel)	*N* = 409 *Enterobacterales*; *n* = 21 *P. aeruginosa*	EA =87% for *Enterobacterales* EA = 100% for *P. aeruginosa* CA = 98% for *Enterobacterales* (ME = 6, VME = 4) CA = 100% for *P. aeruginosa*	[130]
MER-VAB	Etest (bioMèrieux)	*n* = 120 CR-*Enterobacterales*	EA = 82% CA = 95% (ME = 2; VME = 1)	[131]
MER-VAB	MIC test Strip (Liofilcheme)	*n* = 120 CR ^c^-*Enterobacterales*	EA = 48% CA = 90% (ME = 7; VME = 0)	[131]
MER-VAB	DD ^d^ (20/10 µg disk content, Becton, Dikinson and Company)	*n* = 120 CR ^c^-*Enterobacterales*	CA = 90% (ME = 4; VME = 0)	[131]
MER-VAB	Etest (bioMérieux)	*n* = 629 *Enterobacterales* *n* = 163 *P. aeurginosa*	EA = 92% for *Enterobacterales* EA = 93% for *P. aeruginosa* CA = 99% for *Enterobacterales* (ME = 1; VME = 2) CA = 97% for *P. aeruginosa* (ME = 4; VME = 0)	[132]
MER-VAB	Vitek 2 system (bioMérieux)	*N* = 449 *Enterobacterales* *n* = 77 *P. aeruginosa*	EA = 98% for *Enterobacterales* EA = 92.2% for *P.aeruginosa* CA = 99% for *Enterobacterales* (ME = 3%; VME = 0.2%) CA = 97% for *P. aeruginosa* (ME = 3%; VME = 0)	[133]
IMI-REL	Etest (bioMérieux)	*n* = 297 Gram-negatives (*n* = 272 *Enterobacterales*; *n* = 25 *P. aeruginosa*)	EA = 90% CA = 96% (ME = 0; VME = 0)	[134]
IMI-REL	MIC Test Strip (Liofilcheme)	*n* = 297 Gram-negatives (*n* = 272 *Enterobacterales*; *n* = 25 *P. aeruginosa*)	EA = 85% CA = 97% (ME = 0; VME = 0)	[134]
IMI-REL	DD ^d^ (10/25 µg disk content, MAST Group)	*n* = 297 Gram-negatives (*n* = 272 *Enterobacterales*; *n* = 25 *P. aeruginosa*)	CA = 74% (ME = 6; VME = 0)	[134]

^a^ EA: Essential agreement; CA: Categorical Agreement; ME: Major Errors; VME = Very Major Errors. For ME and VME the number or % of errors were reported; ^b^ CP: carbapenemase producers; ^c^ CR: carbapenem resistant. ^d^ DD: disk diffusion.

## Data Availability

All material all available upon request.

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
