# Peer review of "Resistance to Ceftazidime/Avibactam, Meropenem/Vaborbactam and Imipenem/Relebactam in Gram-Negative MDR Bacilli: Molecular Mechanisms and Susceptibility Testing"

_antibiotics, 2022, doi:10.3390/antibiotics11050628_

Round 1

Reviewer 1 Report

Dear authors,

Thank you for your submission.

After careful reading, the overall review focuses on MDR-pathogenic bacilli using molecular and susceptibility tests. However, there are some comments all are in yellow highlighted colors) that can further improve the current form of this review esp. provide infographics under each subheading to increase the visibility and scientific values. Also, authors can provide summarized paragraphs of the benefits in each section which give better understanding of the contents.  

Sincerely,

Author Response

We thanks the reviewer for his comments. We modified the manuscript accordingly to the Reviewer’ suggestions. Also, although reviewer suggested to add infographics under each subheading and a summarize of each paragraphs we choice to maintain the manuscript in the current classical form.

Reviewer 2 Report

This review deals with novel combinations of β-lactams/β-lactamase inhibitors from a mechanistic (mechanism of action and resistance) and clinical (efficacy, susceptibility testing) point of view. The text is well structured and the content is highly relevant for the journal and the special issue on Multi-Drug Resistant Gram-Negative Infections. I have a few minor comments:

The text contains some grammatical and stylistic mistake and should be thoroughly proofread. In particular, the authors should avoid repetition in sentences or paragraphs (for instance, “particular” lines 22-23; “grouped/group” lines 45, “at the same time” lines 46&52, etc). 

The introduction should include a sentence reminding the mechanism of action of β-lactam.

Line 49, “CPE” should be introduced

Line 66, The expression “non-beta-lactam β-lactamase inhibitor” should be more explicit at first occurrence. For instance “β-lactamase inhibitor without beta-lactam motif”

Line 69, “and” is missing.

Line 70, “multi-resistant” should be deleted.

Line 95 “ESβLs” was already introduced before.

Line 116, “MIC” needs to be introduced.

Line 159, “8 patients”. The authors should add the percentage, or the total number of patients included in the study.

Line 215, “to be associated” should be deleted

Part 3 contains a lot of numerical information that could be restricted to Table 2 for clarity.

Line 490, a reference is missing to know who the “authors” (line 489) are.

Units (in particular liters) should be written homogeneously throughout the text.

Adding a graphical figure summarizing important take-home messages would increase the visibility of the review.

Author Response

The text contains some grammatical and stylistic mistake and should be thoroughly proofread. In particular, the authors should avoid repetition in sentences or paragraphs (for instance, “particular” lines 22-23; “grouped/group” lines 45, “at the same time” lines 46&52, etc).

Authors’ reply and amendments: We thanks reviewer for his comment. We modified the manuscript accordingly to the Reviewer comments

The introduction should include a sentence reminding the mechanism of action of β-lactam.

Authors’ reply and amendments: A sentence regarding the mechanism of action of β-lactam was added in the introduction section, as requested.

Line 49, “CPE” should be introduced

Authors’ reply and amendments: The term was modified, as requested

Line 66, The expression “non-beta-lactam β-lactamase inhibitor” should be more explicit at first occurrence. For instance “β-lactamase inhibitor without beta-lactam motif”

Authors’ reply and amendments: The sentence was modified, as requested

Line 69, “and” is missing.

Authors’ reply and amendments: The term was added, as requested

Line 70, “multi-resistant” should be deleted.

Authors’ reply and amendments: The term was removed, as requested

Line 95 “ESβLs” was already introduced before.

Authors’ reply and amendments: The term was removed, as requested

Line 116, “MIC” needs to be introduced.

Authors’ reply and amendments: The term was extended, as requested

Line 159, “8 patients”. The authors should add the percentage, or the total number of patients included in the study.

Authors’ reply and amendments: The number of patients was added, as requested

Line 215, “to be associated” should be deleted

Authors’ reply and amendments: The term was removed, as requested

Part 3 contains a lot of numerical information that could be restricted to Table 2 for clarity.

Authors’ reply and amendments: We thanks reviewer for his comment. However we opted to maintain numerical information in the text for clarity

Line 490, a reference is missing to know who the “authors” (line 489) are.

Authors’ reply and amendments: The reference was added, as requested

Units (in particular liters) should be written homogeneously throughout the text.

Authors’ reply and amendments: The Units were modified as requested

Adding a graphical figure summarizing important take-home messages would increase the visibility of the review.

Authors’ reply and amendments: We thanks reviewer for his comment. However we opted to not to add graphical abstract